# The Minimum AC Signal Model of Bipolar Transistor in Amplification Region for Weak Signal Detection

**DOI:** 10.3390/s21217102

**Published:** 2021-10-26

**Authors:** Lidong Huang, Qiuyan Miao, Xiruo Su, Bin Wu, Kaichen Song

**Affiliations:** 1Ocean College, Zhejiang University, Hangzhou 310058, China; lidonghuang@zju.edu.cn; 2College of Biomedical Engineering and Instrument Science, Zhejiang University, Hangzhou 310058, China; 11715040@zju.edu.cn (Q.M.); xiruo_su@zju.edu.cn (X.S.); 3School of Aeronautics and Astronautics, Zhejiang University, Hangzhou 310058, China; kcsong@zju.edu.cn

**Keywords:** the minimum signal model, the uncertainty principle, carrier diffusion principle, current amplification effect

## Abstract

This paper presents a minimum signal model via the AC small-signal model and the uncertainty principle, which reveals the minimum AC signal that can be amplified by a bipolar transistor. The Ebers—Moll model (EM3) can describe the small signal amplification process, but it is difficult to define the minimum amplifiable signal of the bipolar transistor. In this study, the correspondence relationship between the non-equilibrium carrier and the electric injection is proved, and the relationship between the life of the non-equilibrium carrier and the measurable signal is proposed by the uncertainty principle. Next, the limit of perceived minimum voltage is also derived in this paper. Then, combining with EM3 model, the minimum AC signal model of bipolar transistor is presented to calculate the minimum voltage signal of bipolar transistor that can be amplified. Finally, a number of the simulation and experiment results show that when the minimum signal in the model is used as input, the carrier concentration of the bipolar transistor does not change and the base electrode cannot perceive the signal, which verifies the validity of the minimum AC signal model.

## 1. Introduction

Nowadays, bipolar transistor amplifiers are widely used in weak signal detection, especially for small AC signals [1]. To achieve weak signal detection, the bipolar transistor amplifier requires lower self noise and better performance. The semiconductor material is the most important factor that determines the performance of the amplifier circuit [2]. On the one hand, a large number of experts and researchers use silicon and germanium with different doping concentrations to improve the amplifying circuit’s ability of perceiving weak signals, while some people use semiconductor materials with different elements to reduce self-noise [3,4]. On the other hand, a lot of theoretical models of bipolar transistor have also been proposed and applied to the internal self-generating capacitance and self-generating conductance of semiconductors, including Ebers–Moll model (EM1), EM2, EM3 and Gummel–Poon model (GP) [5]. These studies have greatly promoted the development of transistor amplifier circuit and improved the weak signal detection ability.

Although these small-signal models can be used to describe the operation of bipolar transistor well, it seems that any small signal can be amplified by bipolar transistor [6]. An important question here is what is the minimum signal limit that a triode amplifier can amplify. If this problem is not solved, it is impossible to know what the small signal limit of bipolar transistor amplifier is, which is a big theoretical obstacle for weak signal detection. So far, weak signal detection has been widely used in underwater acoustic detection, target recognition, geophysical exploration and other important scientific fields [7,8,9]. Weak signal detection is based on bipolar transistor amplifier circuit, so the ability of bipolar transistor to detect weak signal is very important [10]. In other words, the most critical problem is how to determine the minimum perceived voltage signal limit of bipolar transistor amplifier circuit. At present, many scientists have done a lot of work and put forward corresponding models to solve this problem, but the problem has not been solved yet [11,12].

So far, there have been related works to promote the performance of the transistor amplifier circuit, mainly including two parts of the transistor manufacturing process and the theoretical calculation model of the transistor [13,14,15]. In terms of semiconductor manufacturing process, T. Maloney et al. [16] have studied the relationship on different materials and the self-noise of semiconductor devices. C. Su et al. [17,18] have developed the gate all-purpose junction less transistor with heavily doped polycrystalline silicon nanowire channel via using a new process, and have studied the current-voltage characteristics associated with the device. Su et al. [19] have designed a silicon-based arch gate omnipotence (GAA) tunnel field effect transistor (TET) and analyzed its current characteristics. With the development of microelectronics, Li et al. [20] find that graphene material has a high carrier mobility, which has a significant advantage in improving semiconductor sensing ability. For the theoretical calculation model of transistor, Ebors and Moll propose a semiconductor equivalent circuit model (EM1) for nonlinear DC analysis [21]. On the basis of the first generation model, considering the effect of nonlinear charge storage and series resistance, the second generation small signal model (EM2) is proposed, which has higher simulation accuracy and faster calculation speed [22]. With the continuous improvement of the calculation model, considering the secondary effects of bipolar transistor amplifier circuit (base region width modulation effect, base region broadening effect, and temperature effect), the third generation of small signal model is put forward to calculate the AC small signal, which has higher accuracy [23]. From the micro-perspective to explore the characteristics of transistor amplifier signal, Gummel-Poon model (G-P) is proposed to establish the connection between the device performance material, structure, and the process parameters [24]. G-P model has three advantages: first, it can directly relate the electrical characteristics of the device to the multi-subcharge in the base region. Second, the small signal problem can be effectively dealt with by its mathematical model [25]. Third, many physical effects can be taken into account by the relationship between the charge of the majority carrier and the bias voltage in the base region [26]. For the detection of weak signal, the main concern is how to amplify the AC small signal. The G-P model is proposed to calculate process of amplifying a small AC signal [27]. These research results greatly promote the development of weak signal detection [28,29]. However, the research on the small signal limit of transistor amplifier is still an unsolved problem, and few scholars and experts have done research and work in this field.

In order to explore the limits of transistor amplifier size signals, several main objectives need to be accomplished in this study. First, we need to prove the equivalent correspondence between the AC small signal and the non-equilibrium carrier. Then, on this basis, the relationship between the life of non-equilibrium carrier and semiconductor material is given in this paper. Next, the minimum limit model is established by combining the Heisenberg uncertainty principle with the EM3 model. At last, the validity of the model is verified by simulating the effective indexes of the triode model, including the changes of internal carriers and base current. The research of this paper solves the problem of the minimum signal limit of the bipolar transistor in the amplification region, which is of great significance for weak signal detection.

The rest of this paper is organized as follows. Section 2 presents the minimum AC signal model of bipolar transistors, which includes the proof of the relationship between the non-equilibrium carrier concentration and the injection voltage, the limit sensing voltage signal model of transistor amplifier, and the improved EM3 model. Section 3 carries out a lot of simulation and completes the analysis in the discussion, which verifies the effectiveness of the minimum AC signal model proposed in this paper. The conclusion and future researches are presented in Section 4.

## 2. The Minimum AC Signal Model

In this section, the paper proposes a minimum AC signal model to calculate the bipolar transistors limit of small signal, which is divided into three parts. The theory of the relationship between non-equilibrium carriers and applied voltage is proposed in first part. The second part presents the limit sensing voltage signal theory of transistor amplifier. The last part improves the EM3 model for bipolar transistor calculation of limit small signal.

### 2.1. The Non-Equilibrium Carrier Theory

In weak signal measurement, bipolar transistors are mainly used to amplify small AC signals [30,31,32]. For the triode amplifier circuit, DC bigoted voltage plus AC small signal is equivalent to the periodic micro-variation of DC voltage. ΔV denotes the voltage change applied to the semiconductor, Δr denotes the change in the resistance inside the semiconductor, and Δσ represents the change in the conductivity of semiconductor. When a bipolar transistor detects a weak signal, the base conductivity of the bipolar transistor changes slightly. At this point, the conductivity of the semiconductor is as follows:(1)$$\sigma =\sigma_{0}+\Delta \sigma \left(\Delta \sigma <<\sigma_{0}\right)$$
where σ denotes the conductivity after the change and $\sigma_{0}$ denotes the conductivity of the semiconductor at equilibrium.

As the conductivity changes, the resistivity of the semiconductor also changes accordingly, as follows:(2)$$\Delta \rho =\frac{1}{\sigma }-\frac{1}{\sigma_{0}}\approx -\frac{\Delta \sigma }{\sigma_{0}^{2}}\propto \Delta \sigma$$

Similarly, the semiconductor resistance changes as follows:(3)$$\Delta r=\frac{\Delta \rho l}{s}\approx -\frac{l\Delta \sigma }{s\sigma_{0}^{2}}\propto \Delta \rho \propto \Delta \sigma$$
where *l* and *s* denote the cross-sectional area and length of the semiconductor, respectively.

From (1)–(3), the applied weak voltage change ΔV is proportional to the semiconductor resistivity Δr. The following relationship can be obtained as follows:(4)ΔV∝Δσ∝Δρ∝Δr

Combining the theory of non-equilibrium carrier diffusion inside the semiconductor [33], the relationship between electronics, holes and conductivity is as follows:(5)$$\Delta \sigma =\Delta nq\mu_{n}+\Delta pq\mu_{p}=\Delta pq\left(\mu_{n}+\mu_{p}\right)\propto \Delta p$$
where Δn, Δp, $\mu_{n}$ and $\mu_{p}$ denote the electron concentration change, the hole concentration change, the electron mobility and the hole mobility, respectively. *q* denotes the electron charge quantity constant.

By combining (4) and (5), the relation between the weak voltage variation of semiconductor and the carrier can be obtained as follows:(6)ΔV∝Δp

In general, the voltage change across the semiconductor is equivalent to the carrier change inside the semiconductor. This relationship provides a theoretical foundation for the establishment of the limiting weak voltage signal model in the next part.

### 2.2. The Limit Small Voltage Signal Theory

From the above conclusion, we know the relationship between the change value of voltage and the change of non-equilibrium carrier concentration. In [34,35], we know that the lifetime of the non-equilibrium carrier can be calculated by the experiment of measuring the change of the concentration of the non-equilibrium carrier. According to the Heisenberg uncertainty principle [36], when the product of its lifetime and energy needs to be greater than the Heisenberg constant, a non-equilibrium carrier to have an effect in a semiconductor, as follows: (7)$$\left.\begin{array}{cc} \; & \Delta E\Delta t\geq \frac{h}{4\pi }\\ \; & \Delta E=q\Delta V\\ \; & \Delta t=\tau \end{array}\right\}.⟺q\Delta V\tau \geq \frac{h}{4\pi }$$
where τ denotes the lifetime of the non-equilibrium semiconductor carrier and *h* denotes Planck constant.

Through (7), the following relation can be obtained as follows:(8)$$\Delta V\geq \frac{h}{4\pi q\tau }$$

From (8), we can draw the corresponding conclusion that when the applied voltage small change is greater than ΔV, the small change can be detected by the semiconductor.

### 2.3. The Minimum Signal Model

In the bipolar transistor amplifier circuit, the base induced AC voltage amplitude should satisfy the following relationship:(9)$$|V_{min}\left|\geq |\Delta V\right|e^{jwt}=\frac{h}{4\pi q\tau }e^{jwt}$$
where *w* denotes frequency of AC voltage and *t* denotes time.

When the bipolar transistor amplifying circuit is in the amplifying region, the total voltage on the base is the DC bias voltage plus the AC small signal, as follows:(10)$$V_{total}=V_{0}+V_{min}\left(t\right)=V_{0}+\frac{h}{4\pi q\tau }e^{jwt}$$
where $V_{0}$ denotes the DC bias voltage.

According to the carrier diffusion theory in the semiconductor [37], the relationship between the change of the hole diffusion concentration $P_{n}$ and the total voltage $V_{total}$ is as follows:(11)$$\begin{array}{cc} P_{n} & =P_{n_{0}}exp\left(\frac{-q(V_{bi}-V_{total})}{KT}\right)=P_{n_{0}}exp\left(\frac{-q(V_{bi}-V_{0}-V_{min}\left(t\right))}{KT}\right)\\ \; & =P_{n_{0}}exp\left(\frac{-qV_{bi}}{KT}\right)exp\left(\frac{qV_{0}}{KT}\right)exp\left(\frac{qV_{min}\left(t\right)}{KT}\right)=P_{dc}exp\left(\frac{qV_{min}\left(t\right)}{KT}\right)\\ \; & =P_{dc}exp\left(\frac{V_{min}\left(t\right)}{KT/q}\right)=P_{dc}exp\left(\frac{V_{min}\left(t\right)}{v_{T}}\right) \end{array}$$
where $V_{bi}$ denotes the barrier voltage inside the bipolar transistor, $P_{n_{0}}$ denotes the intrinsic hole concentration of a semiconductor, *K* denotes Boltzmann constant and *T* denotes the temperature. The thermal voltage of a semiconductor is defined as $v_{T}=KT/q$ and the hole concentration constant at equilibrium is defined as $P_{dc}=P_{n_{0}}exp\left(-qV_{bi}/KT\right)exp\left(qV_{0}/KT\right)$.

According to the semiconductor carrier diffusion theory [38], the hole diffusion equation in the semiconductor under equilibrium state is
(12)$$D_{p}\frac{\partial^{2}\left(\delta_{P_{n}}\right)}{\partial x^{2}}-\frac{\delta_{P_{n}}}{\tau_{p}}=\frac{\delta_{P_{n}}}{t}$$
where $D_{p}$ is the rate of hole formation, $\delta_{P_{n}}$ is the change in hole concentration and $\tau_{p}$ is the lifetime of the hole.

In the case of applied voltage, the solution of the diffusion Equation (Equation 12) should be as follows:(13)$$\delta_{P_{n}}=\delta_{P_{0}}+P_{1}\left(x\right)e^{jwt}$$
where $\delta_{P_{0}}$ is the change of hole under DC voltage and $P_{1}\left(x\right)e^{jwt}$ is the change of hole under small AC voltage.

Combining (12) and (13), the semiconductor hole diffusion equation under the applied electric field can be obtained as follows:(14)$$D_{p}\left[\frac{\partial^{2}\left(\delta_{P_{0}\left(x\right)}\right)}{\partial x^{2}}+\frac{\partial^{2}P_{1}\left(x\right)}{\partial x^{2}}e^{jwt}\right]-\frac{\delta_{P_{0}\left(x\right)}+P_{1}\left(x\right)e^{jwt}}{\tau_{P}}=jwP_{1}\left(x\right)e^{jwt}$$

According to the diffusion properties of semiconductor carriers [39], the semiconductor diffusion equation needs to satisfy three boundary conditions as follows:(15)$$D_{p}\frac{\partial^{2}\left(\delta_{P_{0}}\left(x\right)\right)}{\partial x^{2}}-\frac{\delta P_{0}\left(x\right)}{\tau_{P}}=0$$
(16)$$D_{p}\frac{\partial^{2}P_{1}\left(x\right)}{\partial x^{2}}-\frac{P_{1}\left(x\right)}{\tau_{P}}-jwP_{1}\left(x\right)=0$$
(17)$$P_{1}\left(x\to +\infty \right)=0$$

Through (14)–(17), the AC solution of the hole diffusion equation can be obtained as follows:(18)$$P_{1}\left(x\to 0\right)=P_{dc}\left(\frac{V_{min}\left(t\right)}{v_{T}}\right)$$

Therefore, the hole current density and its current at a small AC voltage are as follows:(19)$${\tilde{J}}_{P}\left(t\right)=qD_{P}P_{dc}\sqrt{\frac{1+jw\tau_{P}}{D_{P}\tau_{P}}}\left(\frac{V_{min}\left(t\right)}{v_{T}}\right)$$
and
(20)$${\tilde{I}}_{P}\left(t\right)=qAD_{P}P_{dc}\sqrt{\frac{1+jw\tau_{P}}{D_{P}\tau_{P}}}\left(\frac{V_{min}\left(t\right)}{v_{T}}\right)$$
where *A* is the cross-sectional area of the base of a bipolar transistor.

Similarly, from the carrier diffusion theory in semiconductor, the relation between the change of electron diffusion concentration $n_{P}$ and the total voltage $V_{total}$ can be obtained as follows:(21)$$\begin{array}{cc} n_{P} & =n_{P_{0}}exp\left(\frac{-q(V_{bi}-V_{total})}{KT}\right)=n_{P_{0}}exp\left(\frac{-q(V_{bi}-V_{0}-V_{min}\left(t\right))}{KT}\right)\\ \; & =n_{P_{0}}exp\left(\frac{-qV_{bi}}{KT}\right)exp\left(\frac{qV_{0}}{KT}\right)exp\left(\frac{qV_{min}\left(t\right)}{KT}\right)=n_{dc}exp\left(\frac{qV_{min}\left(t\right)}{KT}\right)\\ \; & =n_{dc}exp\left(\frac{V_{min}\left(t\right)}{KT/q}\right)=n_{dc}exp\left(\frac{V_{min}\left(t\right)}{v_{T}}\right) \end{array}$$

According to the semiconductor carrier diffusion theory [40], the electron diffusion equation under the semiconductor equilibrium state is as follows:(22)$$D_{n}\frac{\partial^{2}\left(\delta_{n_{P}}\right)}{\partial x^{2}}-\frac{\delta_{n_{P}}}{\tau_{n}}=\frac{\delta_{n_{P}}}{t}$$
where $D_{n}$ is the rate of electron formation, $\delta_{n_{P}}$ is the change in electron concentration, and $\tau_{n}$ is the lifetime of the electron.

Similarly, under applied voltage, the solution of the diffusion Equation (22) should be as follows:(23)$$\delta_{n_{P}}=\delta_{n_{0}}+n_{1}\left(x\right)e^{jwt}$$
where $\delta_{n_{0}}$ is the change of electron under DC voltage and $n_{1}\left(x\right)e^{jwt}$ is the change of hole under small AC voltage.

Combining (22) and (23), the semiconductor electron diffusion equation under the applied electric field can be obtained as follows:(24)$$\begin{array}{c} D_{n}\left[\frac{\partial^{2}\left(\delta_{n_{0}\left(x\right)}\right)}{\partial x^{2}}+\frac{\partial^{2}n_{1}\left(x\right)}{\partial x^{2}}e^{jwt}\right]-\frac{\delta_{n_{0}\left(x\right)}+n_{1}\left(x\right)e^{jwt}}{\tau_{n}}=jwn_{1}\left(x\right)e^{jwt} \end{array}$$

In the same way, according to the diffusion properties of semiconductor carriers, the semiconductor diffusion equation needs to satisfy three boundary conditions as follows:(25)$$D_{n}\frac{\partial^{2}\left(\delta_{n_{0}}\left(x\right)\right)}{\partial x^{2}}-\frac{\delta n_{0}\left(x\right)}{\tau_{n}}=0$$
(26)$$D_{n}\frac{\partial^{2}n_{1}\left(x\right)}{\partial x^{2}}-\frac{n_{1}\left(x\right)}{\tau_{n}}-jwn_{1}\left(x\right)=0$$
(27)$$n_{1}\left(x\to -\infty \right)=0$$

Through (24)–(27), the AC solution of the electron diffusion equation can be obtained as follows:(28)$$n_{1}\left(x\to 0\right)=n_{dc}\left(\frac{V_{min}\left(t\right)}{v_{T}}\right)$$

Therefore, the electron current density and its current at a small AC voltage are as follows:(29)$${\tilde{J}}_{n}\left(t\right)=qD_{n}n_{dc}\sqrt{\frac{1+jw\tau_{n}}{D_{n}\tau_{n}}}\left(\frac{V_{min}\left(t\right)}{v_{T}}\right)$$
and
(30)$${\tilde{I}}_{n}\left(t\right)=qAD_{n}n_{dc}\sqrt{\frac{1+jw\tau_{n}}{D_{n}\tau_{n}}}\left(\frac{V_{min}\left(t\right)}{v_{T}}\right)$$

Therefore, the minimum current caused by the minimum AC voltage is as follows:(31)$$\begin{array}{c} {\tilde{I}}_{min}\left(t\right)={\tilde{I}}_{n}+{\tilde{I}}_{P}=qA\left(\frac{V_{min}\left(t\right)}{v_{T}}\right)\left(D_{n}n_{dc}\sqrt{\frac{1+jw\tau_{n}}{D_{n}\tau_{n}}}+D_{P}P_{dc}\sqrt{\frac{1+jw\tau_{P}}{D_{P}\tau_{P}}}\right) \end{array}$$

In total, the limit changes of resistance, conductivity, carrier concentration, and current, which are caused by the small signal of the applied AC voltage, are shown below.
(32)$$V_{min}\left(t\right)\propto \Delta r_{min}\left(t\right)\propto \Delta \rho_{min}\left(t\right)\propto \Delta \sigma_{min}\left(t\right)\propto \Delta P_{min}\left(t\right)\propto I_{min}\left(t\right)$$

The minimum sensing signal model of the bipolar transistor is described in three parts, which explain the whole process of carrier change and current diffusion caused by the minimum limit voltage.

## 3. Simulation and Experiment

### 3.1. The Simulation of Model

In this section, a number of simulations are designed to verify the minimum AC signal model proposed in this paper. We design an NPN bipolar transistor with a parameter of 1.25 μm × 1.5 μm × 1 μm, as shown in Figure 1a. The initial potential distribution of the intrinsic semiconductor under the equilibrium state is shown in Figure 1b. In this simulation, bipolar transistors of different semiconductor materials are used to calculate the corresponding base carrier distribution and its base current characteristics. First, the minimum perceived voltage of semiconductor is calculated according to the carrier life of different semiconductor materials. Then, when the minimum voltage is used as the base drive, the change in the carrier distribution concentration of the base is calculated. Finally, the corresponding base current response is calculated by the minimum AC voltage signal model.

This section uses different semiconductor materials to simulate, including silicon, germanium, gallium arsenide and different doping concentrations of germanium. When T=300 K and other environmental factors are in ideal state, the corresponding carrier lifetime and their minimum perceived voltage of different semiconductor materials are shown in the Table 1.

According to the minimum signal model, we get the theoretical minimum voltage of different semiconductor materials and use it as the input of transistor base. This paper focuses on the detection of weak signals by bipolar transistors, so the transistors are in the amplification region (The base is applied with DC bias voltage). We can get the potential distribution, carrier concentration variation and the corresponding voltage-current characteristics of the bipolar transistor by inputting a sinusoidal voltage signal with the amplitude of $V_{min}$ and the frequency of 73 Hz at the base. The effectiveness of the minimum signal model proposed by this paper can be verified by analyzing these results. The potential distribution, carrier concentration variation and the corresponding voltage-current characteristics of the different bipolar transistor are shown in Figure 2, Figure 3, Figure 4, Figure 5 and Figure 6.

Figure 2, Figure 3, Figure 4, Figure 5 and Figure 6 show the potential distribution, carrier concentration changing with time, and current response of base of bipolar transistor with different carrier life. First, Figure 2a, Figure 3a, Figure 4a, Figure 5a and Figure 6a show the potential diagram of semiconductor with silicon (Si) doping concentration of $1.5\times 10^{10}$ cm${}^{-3}$, germanium (Ge) doping concentration of $2.4\times 10^{13}$ cm${}^{-3}$, gallium arsenide (GaAs) doping concentration of $1.8\times 10^{6}$ cm${}^{-3}$, impurity germanium (Ge-1) doping concentration of $1.2\times 10^{13}$ cm${}^{-3}$ and impurity germanium (Ge-2) doping concentration of $4.8\times 10^{13}$ cm${}^{-3}$, respectively. It is observed that the potential of the base does not fluctuate significantly.

Next, Figure 2b, Figure 3b, Figure 4b, Figure 5b and Figure 6b show the base carrier concentration change of the different bipolar transistors in one second. At respective minimum sensing voltage of different material ($0.6\times 10^{-12}$ V, $0.6\times 10^{-13}$ V, $0.6\times 10^{-7}$ V, $0.3\times 10^{-11}$ V, and $0.1\times 10^{-10}$ V), these corresponding carrier concentration remain around $1.5\times 10^{10}$ cm${}^{-3}$, $2.4\times 10^{13}$ cm${}^{-3}$, $1.8\times 10^{6}$ cm${}^{-3}$, $1.2\times 10^{13}$ cm${}^{-3}$, and $4.8\times 10^{13}$ cm${}^{-3}$, respectively. We can know from the minimum signal model that the change of applied voltage is proportional to the change of carrier concentration. However, the carrier concentration of the base of these different transistors does not change periodically with the applied sinusoidal voltage signal, which indicates that the minimum voltage signal is not perceived by the base of bipolar transistors.

Then, Figure 2c, Figure 3c, Figure 4c, Figure 5c and Figure 6c show the corresponding minimum sinusoidal voltage signal and base current response of different bipolar transistors in one second. From these figures, these small signals applied to the base satisfy the minimum limit voltage of different semiconductor materials and maintain the frequency characteristics, but their respective base current do not produce corresponding frequency response and periodic changes. From Figure 2, Figure 3, Figure 4, Figure 5 and Figure 6c, the base response voltage is almost zero and there is no change in the carrier concentration, which are $1.5\times 10^{10}$, $2.4\times 10^{13}$, $1.8\times 10^{6}$, $1.2\times 10^{13}$ and $4.8\times 10^{13}$. In others words, it shows that the bases of these transistors does not sense the applied sine voltage signal applied by them.

From the potential distribution, carrier concentration changing with time, and voltage response of bases of different bipolar transistor, we can draw a conclusion that when the minimum limit voltage signal is applied to the bipolar transistor, its potential, carrier concentration and response voltage will not change as the sinusoidal voltage signal, which verifies the effectiveness of the minimum signal model. Generally speaking, if the weak voltage signal is less than the minimum limit voltage of the semiconductor, the bipolar transistor amplifier circuit can not detect the signal.

### 3.2. The Experiment of Model

In this part, a number of experiments are designed to verify the minimum limit signal model of bipolar transistor. First, We choose bipolar transistors of materials Si, Ge, GaAs, Impurity-1 Ge and Impurity-2 Ge, respectively, and they are as front-end amplifiers of the signal acquisition circuit. ART-PXI8812-PXI with the sampling accuracy of 24-bits is used as the data acquisition module and its the measurement range and sampling rate are selected as ±1 V and 500 Hz in the experiment, respectively. Similarly, we use AWG5200 as signal generator to generate a sine wave of frequency 73 Hz. Then, we use the signal attenuator of XMA to achieve the amplitude of the input signal at $0.6\times 10^{-7}$ V, $0.3\times 10^{-11}$ V and $0.1\times 10^{-10}$ V respectively. Finally, the experiment is carried out in an ultra-static shielded room. In order to suppress background noise, different acquisition lengths are needed to satisfy the time gain.

In this experiment, the time gain characteristic of Fourier transform is used to judge whether the signal appears on the spectrum. Due to the difference in the amplitude of the input signal, each signal acquisition experiment requires different time gain. The input signal of $0.6\times 10^{-7}$ V, $0.3\times 10^{-11}$ V, and $0.1\times 10^{-10}$ V need the time length of 24 h, 168 h and 96 h, respectively.

Figure 7a shows the input voltage signal and its frequency spectrum. After 24 h of data collection, we can get the signal and its frequency spectrum in Figure 7b. With time gain of 24 h, background noise is suppressed to $0.4\times 10^{-7}$ V. We can observe that there is no signal with amplitude of $0.6\times 10^{-7}$ V and frequency of 73 Hz on the frequency spectrum. That explains that when the amplitude of input voltage signal is $0.6\times 10^{-7}$ V, the bipolar transistor of material GaAs does not sense the input signal.

Figure 8a gives the input voltage signal and its frequency spectrum. With 168 h of data collection, we obtain the signal and its frequency spectrum in Figure 8b. With time gain of 168 h, background noise is suppressed to $0.3\times 10^{-11}$ V. We can observe that there is no signal with amplitude of $0.3\times 10^{-11}$ V and frequency of 73 Hz on the frequency spectrum. That explains that when the amplitude of input voltage signal is $0.3\times 10^{-11}$ V, the bipolar transistor of material GaAs does not sense the input signal.

Figure 9a provides the input voltage signal and its frequency spectrum. After 168 h of data collection, we get the signal and its frequency spectrum in Figure 9b. With time gain of 96 h, background noise is suppressed to $0.1\times 10^{-10}$ V. We can observe that there is no signal with amplitude of $0.1\times 10^{-10}$ V and frequency of 73 Hz on the frequency spectrum. That explains that when the amplitude of input voltage signal is $0.1\times 10^{-10}$ V, the bipolar transistor of material GaAs does not sense the input signal.

In total, from Figure 7, Figure 8 and Figure 9, we can draw a conclusion that when the minimum limit voltage signal is applied to the bipolar transistor, its response voltage signal will not change as the sinusoidal voltage signal, which verifies the effectiveness of the minimum signal model. Generally speaking, if the weak voltage signal is less than the minimum limit voltage of the semiconductor, the bipolar transistor amplifier circuit cannot detect the signal.

## 4. Conclusions

In this paper, a minimum AC voltage signal model is proposed to illustrate the minimum perceptible signal limit of bipolar transistor. This model not only proves the positive proportional relationship between the weak voltage signal and the carrier from the microscopic point of view, but also puts forward a clear minimum limit voltage signal theory of triodes via combining Hessian uncertainty principle and improved EM3 model. Finally, from three aspects of the semiconductor potential distribution—the change of carrier concentration with time, the base voltage response and output response voltage signal, the simulation and experiment of bipolar transistors with different materials is carried out to verify the effectiveness of the model, which is of great significance for weak signal detection.

## Figures and Tables

**Figure 1 sensors-21-07102-f001:**
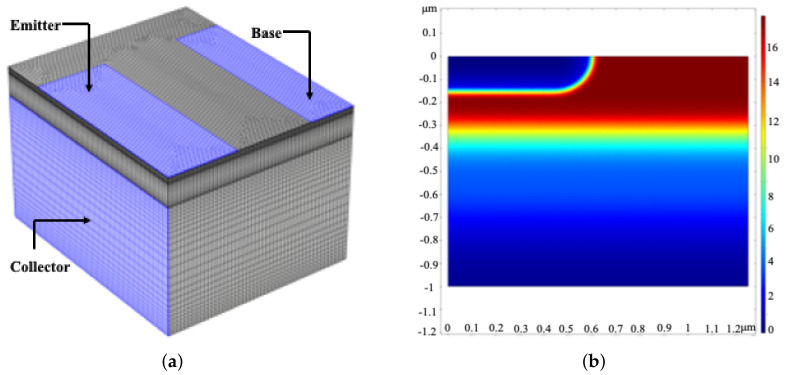
(**a**) The three-dimensional model of NPN bipolar transistor. (**b**) The initial potential distribution of the intrinsic semiconductor. (Two-dimensional section:*x* axis represents the length of the model, *y* axis represents the height of the model. There is no external electric field.)

**Figure 2 sensors-21-07102-f002:**
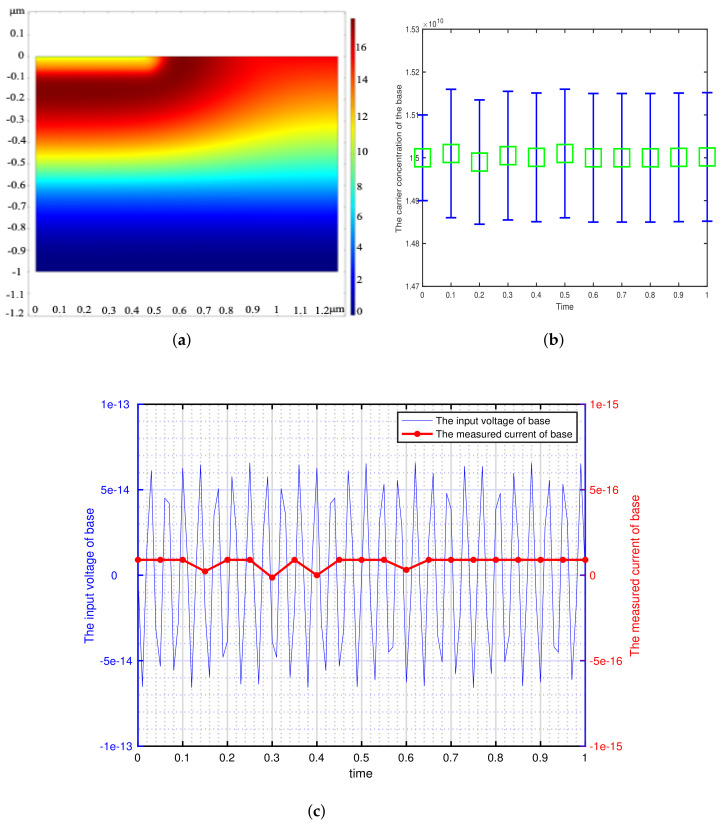
The corresponding sinusoidal voltage signal is applied to the bipolar transistor in the amplification region, which is made of silicon (Si) with doping concentration of $1.5\times 10^{10}$ cm${}^{-3}$ and carrier lifetime of $10^{-3}$ s. (**a**) The potential distribution of the intrinsic semiconductor under sinusoidal small voltage signal. (2-D front view) (**b**) The variation of base carrier concentration with time. (**c**) The input weak sinusoidal voltage signal and corresponding base current response.

**Figure 3 sensors-21-07102-f003:**
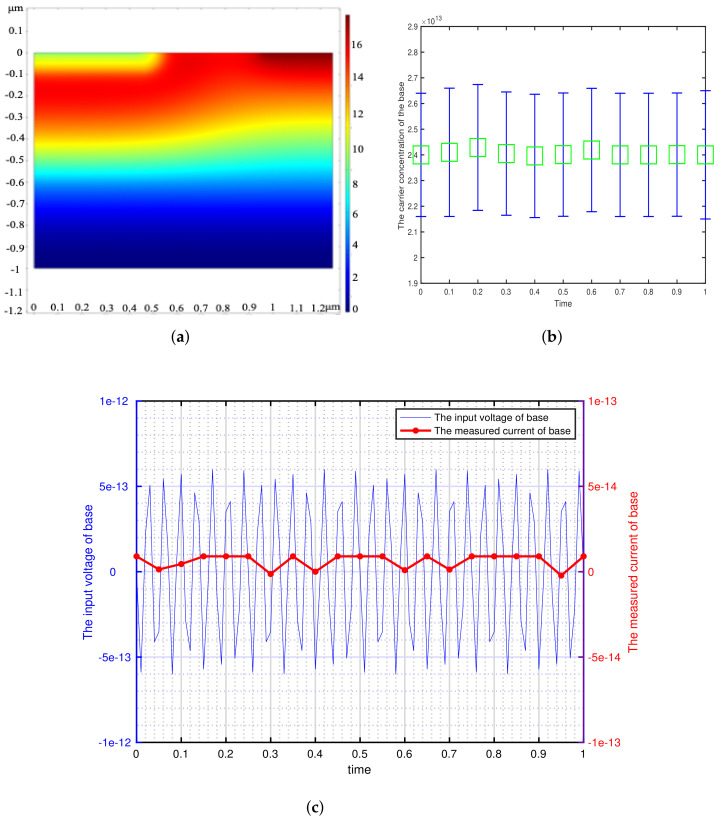
The corresponding sinusoidal voltage signal is applied to the bipolar transistor in the amplification region, which is made of germanium (Ge) with doping concentration of $2.4\times 10^{13}$ cm${}^{-3}$ and carrier lifetime of $10^{-2}$ s. (**a**) The potential distribution of the intrinsic semiconductor under sinusoidal small voltage signal. (2-D front view) (**b**) The variation of base carrier concentration with time. (**c**) The input weak sinusoidal voltage signal and corresponding base current response.

**Figure 4 sensors-21-07102-f004:**
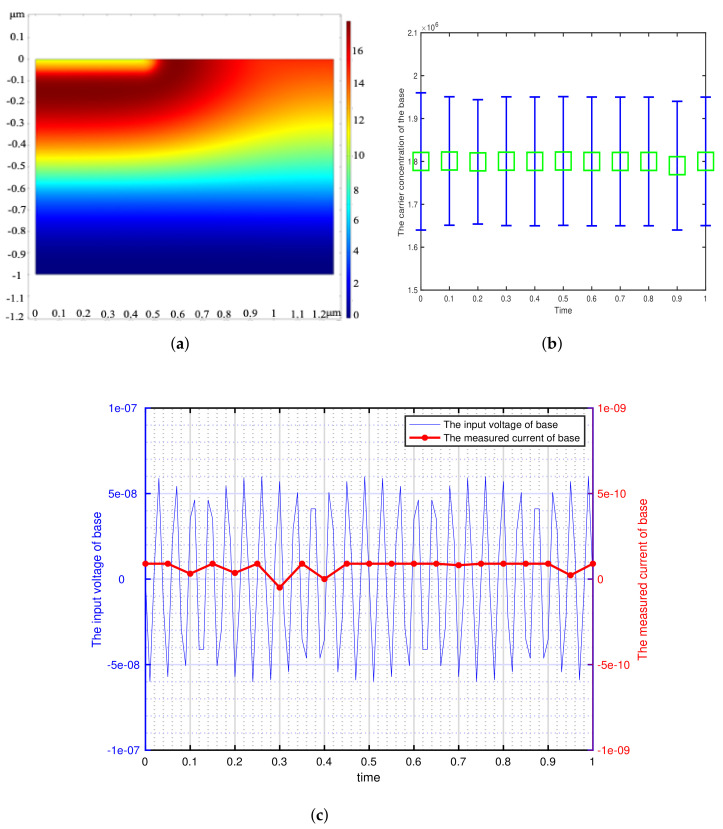
The corresponding sinusoidal voltage signal is applied to the bipolar transistor in the amplification region, which is made of gallium arsenide (GaAs) with doping concentration of $1.8\times 10^{6}$ cm${}^{-3}$ and carrier lifetime of $10^{-8}$ s. (**a**) The potential distribution of the intrinsic semiconductor under sinusoidal small voltage signal. (2-D front view) (**b**) The variation of base carrier concentration with time. (**c**) The input weak sinusoidal voltage signal and corresponding base current response.

**Figure 5 sensors-21-07102-f005:**
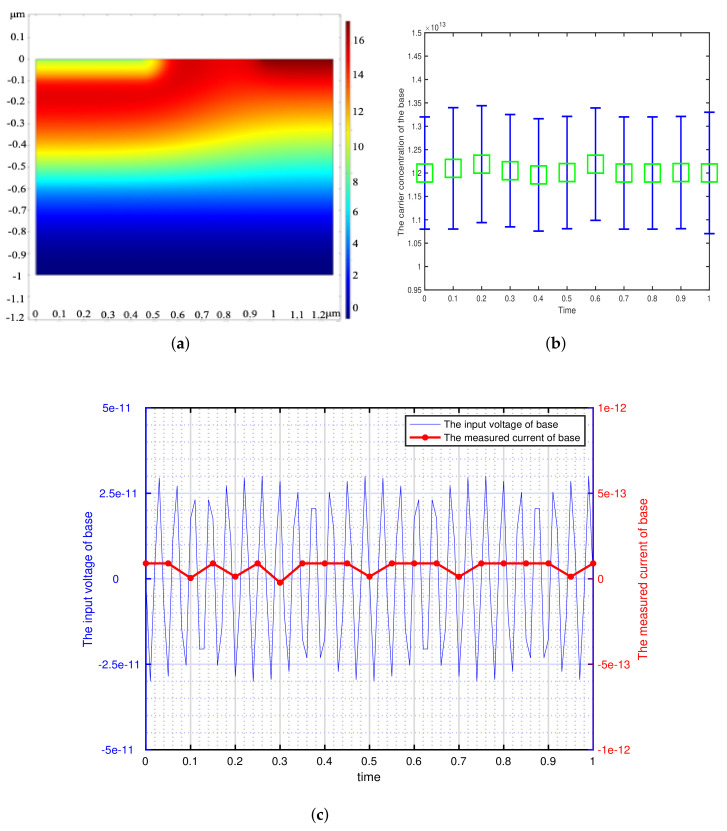
The corresponding sinusoidal voltage signal is applied to the bipolar transistor in the amplification region, which is made of impurity germanium (Ge-1) with doping concentration of $1.2\times 10^{13}$ cm${}^{-3}$ and carrier lifetime of $2\times 10^{-4}$ s. (**a**) The potential distribution of the intrinsic semiconductor under sinusoidal small voltage signal. (2-D front view) (**b**) The variation of base carrier concentration with time. (**c**) The input weak sinusoidal voltage signal and corresponding base current response.

**Figure 6 sensors-21-07102-f006:**
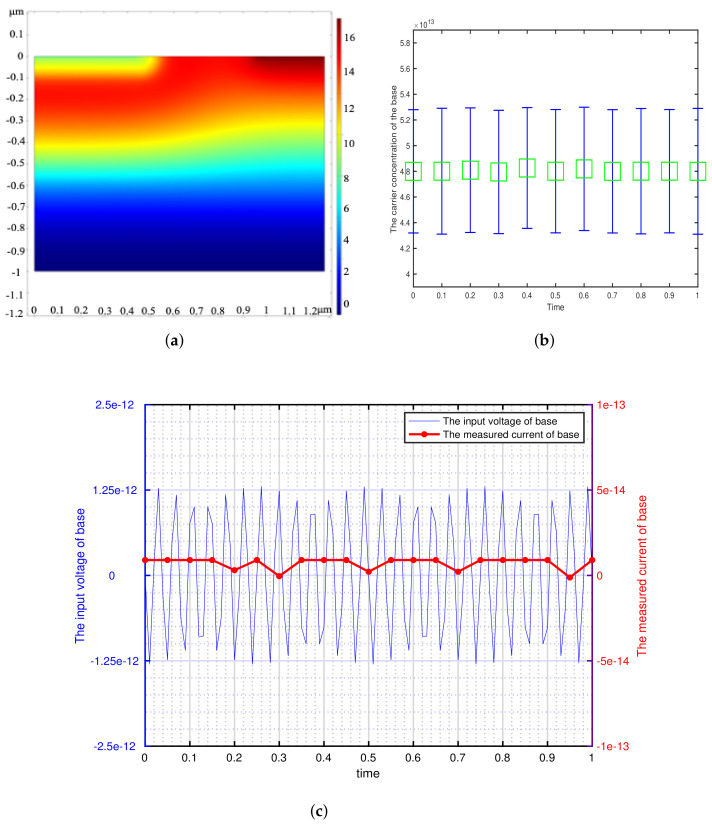
The corresponding sinusoidal voltage signal is applied to the bipolar transistor in the amplification region, which is made of impurity germanium(Ge-2) with doping concentration of $4.8\times 10^{13}$ cm${}^{-3}$ and carrier lifetime of $5\times 10^{-5}$ s. (**a**) The potential distribution of the intrinsic semiconductor under sinusoidal small voltage signal. (2-D front view) (**b**) The variation of base carrier concentration with time. (**c**) The input weak sinusoidal voltage signal and corresponding base current response.

**Figure 7 sensors-21-07102-f007:**
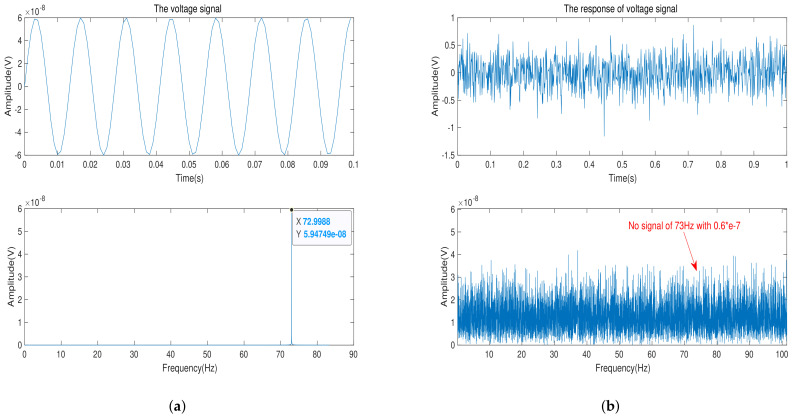
(**a**) The input-voltage signal with amplitude of $0.6\times 10^{-7}$ V and its frequency spectrum. (**b**) The acquisition voltage signal in time-domain and its frequency spectrum after time gain of 24 h.

**Figure 8 sensors-21-07102-f008:**
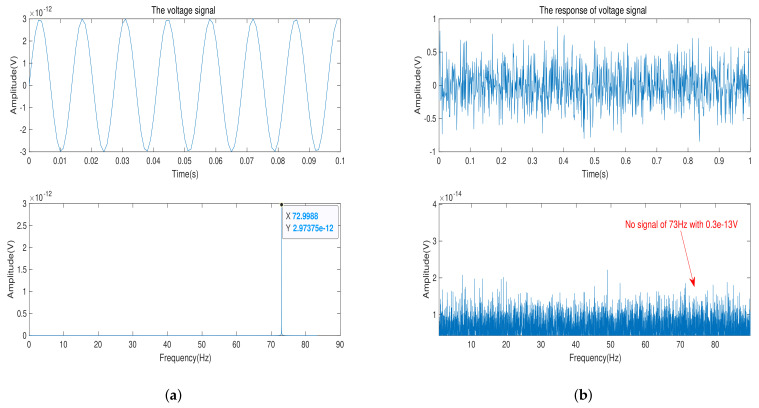
(**a**) The input-voltage signal with amplitude of $0.3\times 10^{-11}$ V and its frequency spectrum. (**b**) The acquisition voltage signal in time-domain and its frequency spectrum after time gain of 168 h.

**Figure 9 sensors-21-07102-f009:**
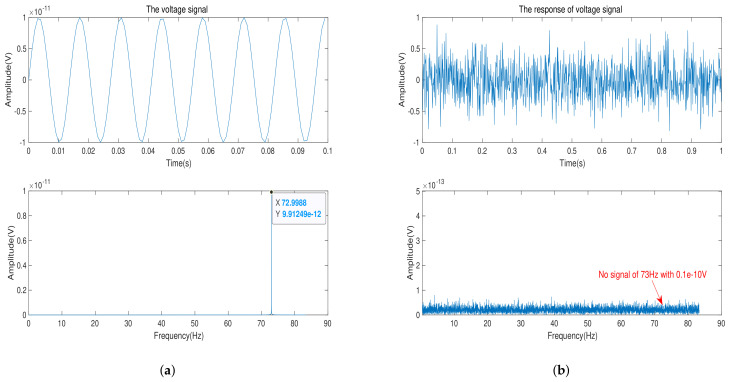
(**a**) The input-voltage signal with amplitude of $0.1\times 10^{-10}$ V and its frequency spectrum. (**b**) The acquisition voltage signal in time-domain and its frequency spectrum after time gain of 96 h.

**Table 1 sensors-21-07102-t001:** The carrier lifetime and their minimum perceived voltage of different semiconductor materials.

The Name of Semiconductor Material	Si	Ge	GaAs	Impurity Ge-1	Impurity Ge-2
The carrier lifetime τ (s)	$10^{-3}$	$10^{-2}$	$10^{-8}$	$2\times 10^{-4}$	$5\times 10^{-5}$
The minimum perceived voltage $V_{min}$ (v)	$0.6\times 10^{-12}$	$0.6\times 10^{-13}$	$0.6\times 10^{-7}$	$0.3\times 10^{-11}$	$0.1\times 10^{-10}$

## Data Availability

The data that support the findings of this study are available from the corresponding author upon reasonable request.

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
