# Peer review of "The Minimum AC Signal Model of Bipolar Transistor in Amplification Region for Weak Signal Detection"

_sensors, 2021, doi:10.3390/s21217102_

Round 1
Reviewer 1 Report
The authors proposed a model for small signal detection by transistor amplifiers, the minimum AC signal model. They discussed the basis on weak signal detection, exposing the previous models and using them to build their own model. The effectiveness of the model proposed was verified by analyzing the results of carrier concentration with time and base current response. Although the conclusion may be interesting, some discussion is needed on them.
In page 2, line 69, should it read “G-P model” instead of “H-P model”?
Could the authors, briefly, justify why, in the microscopic semiconductor theory, the applied weak voltage change ΔV is proportional to the semiconductor resistivity Δr (line 111, page 3)?
In equation (5), why do the electron and hole concentration change are considered equal?
The uncertainty principle stands that ΔE.Δt≥h/4π. Should not that be written in equation (7) instead of q.ΔV.t≥h/2π?
In section “3. Simulation and Discussion”, page 7, the dimensions of the transistor were given but with no identification to which axis they corresponded.
Graphs axis and labels are very hard to read. Could the authors improve them?
Could some graphs be shown in Supplementary Information?
From the analysis of figures 2 to figure 6, label (a), the authors claimed that the potential of the base does not fluctuate significantly. Can they provide a more quantitative analysis?
The results of carrier concentration with time and base current response indicated that the minimum voltage signal is not perceived by the base of bipolar transistors. Did the authors verify the behavior of both parameters with the input potential? Is it possible to observe a phase transition of the system?
Reviewer 2 Report
The theoretical background unfolded in the paper points out main issues that leads the reader to important confusions and misunderstandings.
Concretely, in the title of the paper is clearly stated that the research is about a bipolar transistor model. However, in the introduction paragraph the discussion is focused already on the triode device: ”Nowadays, triode amplifiers are widely used in weak signal detection, especially for small AC signals [1]. To achieve weak signal detection, the triode amplifier requires lower self noise and better performance.” Toward in the second paragraph: “Although these small-signal models can be used to describe the operation of triodes well, it seems that any small signal can be amplified by triodes [6]. An important question here is what is the minimum signal limit that a triode amplifier can amplify.”
Other: “Weak signal detection is based on triode amplifier circuit, so the ability of triode to detect weak signal is very important [10]. In other words, the most critical problem is how to determine the minimum perceived voltage signal limit of triode amplifier circuit.”
Such other similar discussions and references to the triode device may be found often in other paragraphs of the paper but the most confusing is the first sentence of the conclusion paragraph: “In this paper, a minimum AC voltage signal model is proposed to illustrate the minimum perceptible signal limit of triode.” In this circumstance, why the title is “The Minimum AC Signal Model of Bipolar Transistor….”?
The mathematical background and the presented simulation results are only about the bipolar transistor and not about the triode. However, why are mixed in the paper the two very different devices? The triode is an electronic amplifying vacuum tube consisting of three electrodes, the bipolar transistor is a semiconductor-based (silicon or germanium) device both with three terminals.
The presented mathematical background is adequate and fulfills the requirement level for journal publication purposes.
Toward, the assumptions made in the paper are supported only by numerical simulation results. Experimental investigations and results misses at all. Why are not presented experimental measurements? Even the presented simulations (potential distribution, carrier distribution, and input/response signals) looks a little bit weak and cannot be considered fully enough to support to presumption made in the paper (more precisely the detection of perceived minimum voltage level).
As conclusion, considering all the above remarks the paper (with this actual form and content) is not recommended for journal publication.
Round 2
Reviewer 1 Report
Authors have addressed the issues raised by this reviewer. The manuscript can be accepted for publication.
Reviewer 2 Report
The authors performed the requested changes according to the reviewer observations.
The paper now looks suitable for journal publication.